# Assessing the Effects of Environmental Smog Warning Policy on Preventing Traffic Deaths Based on RDD Strategy

**Juan Gao [1], Cheng Ying [2], Liyuan Hu [3], Zixiang Lin [3],* and Hao Xie [4]**

1  School of Political Science and Public Administration, Wuhan University, Wuhan 430072, China; gaojuan33@whu.edu.cn
2  Power China Hubei Electric Engineering Co., Ltd., Wuhan 430040, China; chengy@powerchina-hb.com
3  Institute of Quality Development Strategy, Wuhan University, Wuhan 430072, China; 2021206390014@whu.edu.cn
4  School of International Education, Wuhan University, Wuhan 430072, China; xiehao@whu.edu.cn
*  Correspondence: 2015301750052@whu.edu.cn

**Abstract:** This paper assessed the impacts of environmental smog early-warning signals on road traffic deaths. For an accurate assessment, we used the daily traffic death data from 2016 to 2020 in 295 Chinese cities and constructed a rigorous Regression Discontinuity Design (RDD) strategy to identify the causality and adopted the high-dimensional fixed-effect method to deal with the interference of meteorological factors. The results indicate that light smog and moderate smog early warnings decreased road fatalities by about 3.6% and 4.3%, respectively. Surprisingly, the heavy smog early-warning signal had no significant effect, possibly because of the self-consciousness mechanism instead of the early-warning signal mechanism. Further heterogeneity analysis showed that women drivers, highly-educated drivers, older drivers (over 60 years), two-wheeled vehicle drivers, and drivers on country roads and freeways are more sensitive to smog early-warning signals.

**Keywords:** smog early-warning; environmental policy; road traffic deaths; Regression Discontinuity Design





## 1. Introduction

Traffic deaths account for most of the unnatural deaths of human beings, and about 1.3 million people die from traffic accidents every year worldwide. Investigating the causes of traffic accidents and improving the corresponding prevention policies are crucial to reducing the volume of traffic deaths. Many studies have indicated that in addition to internal subjective factors such as careless driving and external meteorological factors such as rain, snow, and fog, environmental factors such as smog pollution [1] are also important drivers of traffic deaths. Many countries have formulated meteorological warning policies, including heavy fog warnings, high-temperature warnings, etc., and environmental warning policies such as smog pollution warnings. It should also be noted that research in the field of traffic safety includes extensive literature on early-warning policies, but almost all of them focus on meteorological early-warning policies, while environmental early-warning policies are largely ignored.

There are obvious differences between environmental early warnings and meteorological early warnings. Taking a meteorological heavy fog early warning and an environmental smog early warning as examples, the former mainly affects people's vision but does not cause harm to their health, while the latter mainly damages the health of individuals and its influence on vision is much less than the former [2]. Previous studies have proved that environmental smog pollution is more likely to result in traffic deaths by exacerbating drivers' physiological and psychological states to reduce their driving performance, rather than blurring the drivers' sight as the heavy fog does [1], since the most serious extremely

heavy smog early-warning can only cause a visual impact of around 1000 m, which has limited impacts on road driving.

In this paper, we will assess the effects of the environmental smog early warning on road traffic deaths. To accurately evaluate the effects, we need to resolve two technical problems. The first is the reverse causal relationship between smog pollution and traffic. Traffic vehicles are important contributors to the formation of smog pollutants through the emission of particulates. While many scholars have investigated the effect of traffic on smog pollution formation, we assess the impact of smog pollution on traffic in reverse. The second problem is the interference of complex and changeable meteorological factors. The formation of smog pollution and the occurrence of traffic accidents are strongly related to weather conditions such as temperature, humidity, heavy fog, wind, etc. We need to remove the interference of these meteorological factors.

In this paper, we utilized the Regression Discontinuity Design (RDD) strategy and the high-dimensional fixed-effect method to address the issue of reverse causality and meteorological interference. The RDD is a quasi-natural experimental method based on the fundamental idea that smog early-warning signals, which are issued based on different PM2.5 concentrations, are the sole factor contributing to the reduction of traffic deaths within small windows near the breakpoints of the early-warning signals. Meanwhile, the other factors must remain relatively unchanged within that range. The slight rise in PM2.5 concentration only causes the occurrence of early-warning signals, and if they lead to discontinuous changes in the number of traffic deaths, those changes may manifest the effects of early-warning signals on traffic deaths. In the RDD strategy, we can also control the high-dimensional fixed effects of temperature, humidity, fog, rain, snow, wind, and other weather factors by groups, which can strictly exclude the interference of meteorological factors on the estimation.

We collected the records of all traffic death cases from 2016 to 2020 on the Chinese Judicial Document Network and sorted out the number of traffic deaths per day in 295 prefectural cities. According to the PM2.5 concentrations of different smog pollution levels (for light smog, moderate smog, and extremely heavy smog, the corresponding PM2.5 concentration thresholds are 150 μg/m$^3$, 250 μg/m$^3$, and 500 μg/m$^3$, respectively), we constructed RDD models to estimate the effects of different smog pollution early-warning signals on reducing traffic deaths. In addition, we also conducted a heterogeneity analysis of the different driver characteristics, vehicle types, and road locations.

This paper contributes to the existing literature in three aspects. Firstly, this paper contributes to the literature regarding the evaluation of environmental policies. Previous researchers focused on "environmental regulation" policies, including environmental laws, environmental administrative regulations [3], or government environmental initiatives and campaigns, and their impacts on human health, economic growth, enterprise performance, or social welfare, while little literature involved "environmental warning" policies. This article concentrates on the effect of environmental smog early-warning policy on traffic deaths, which can enrich the literature on environmental policy from the perspective of "warning" rather than "regulation" and extend the research of environmental policy to the field of traffic safety as well.

Secondly, this paper reveals a new early-warning policy to prevent traffic deaths from the environmental aspect rather than from the meteorological perspective. Existing studies of traffic safety early warnings have paid attention to severe weather conditions, including heavy fog warnings, rain and snow warnings, typhoon warnings, and high-temperature warnings. However, the effect of environmental early-warning signals has been neglected. It is hoped that the conclusion of this paper will inspire traffic sectors to further enrich traffic safety policies from the environmental aspect.

Thirdly, this paper designs a rigorous econometric method for assessing environmental smog early-warning policies. Scholars usually use panel data with the fixed effects model or Difference-in-Differences methods to evaluate environmental policies, while we design

a stricter RDD strategy based on the PM2.5 concentration thresholds of the smog early warnings which can offer methodological ideas for follow-up research.

The contents of this article are arranged as follows. The second section introduces the policy background and includes a literature review, the third section explains the research strategies and methods, the fourth section shares the main empirical results, the fifth section discusses the results, and the sixth section provides a conclusion.

## 2. Policy Background and Literature Review

### 2.1. Environmental Smog Early-Warning Policy

While China's economy has maintained high-speed growth, it has also brought about increasingly severe environmental pollution problems since the 21st century [4,5]. The Chinese government has carried out a nationwide early-warning environmental policy aiming to inform the public to avoid pollution harm, known as the smog early-warning policy. Small particles in smog pollution (such as PM2.5) can enter people's pulmonary alveolar or even the blood from breathing, inducing a variety of diseases of the human respiratory system and circulatory system, thus increasing their prevalence in the population and mortality, reducing the regional life expectancy [6]. In recent years, some of the literature has pointed out that smog also reduces people's cognitive abilities and behavioral performance, and thus has an invisible impact on economic and social activities based on individual behaviors, such as population migration [7], crime rate, labor productivity, etc. The China Environment Protection Administration (CEPA) began monitoring fine particles in the air (such as PM2.5 and PM10) in 2011 and formulated smog early-warning levels based primarily on PM2.5 concentration to warn everyone to reduce outdoor activities or wear protective masks when going out starting in 2013.

Different from similar meteorological heavy fog warnings, which focus on visual effects, the smog warnings pay more attention to the extent of exposure to health damage from particulate matter. The heavy fog weather warning policy is mainly for traffic drivers, to remind them to drive carefully. The transportation sector usually attaches great importance to such warning signals, and accordingly takes some preventive traffic control measures, such as closing roads and artificially guiding traffic flow [8], to minimize the negative impact of meteorological factors on traffic safety. Compared with the explicit influence of meteorological factors such as heavy fog on traffic safety, smog, as an environmental pollution factor, has a more hidden impact on safe driving, mainly through health effects, other than through the visual effects of heavy fog. The smog warning policy is aimed at the public to remind them to reduce outdoor activities to avoid the harm caused by tiny particulate matter. However, since the warning is not traffic-oriented, the traffic departments usually do not take any traffic control measures when they only receive the smog warning signals [9]. To the best of our knowledge, there seems to be no research on the effect of environmental warning policy on traffic safety.

### 2.2. Smog Pollution and Traffic Safety

Since car driving is a continuous, independent, and long-term process that consumes a lot of human energy and physical strength, it is necessary that the driver have good physiological functions such as vision and hearing, and fine physical capabilities such as concentration, reaction speed, and coordination, as well as a healthy psychological quality and the mental state which can accurately predict the road condition and sensibly evade danger [10]. Heavy smog will significantly reduce the driver's visibility, resulting in delayed response and avoidance, thus increasing the risk of road traffic accidents. However, Sager's [1] study shows that even light smog pollution that is not easy to detect visually, such as the invisible fine particles of PM2.5, will also affect safe driving on the road.

Smog pollution acts covertly on the driver's physiology and mental or cognitive state, and then influences safe driving behaviors. Smog pollution can also cause individual physical discomfort. People breathe in fine particles floating in the air, such as PM2.5, into the pulmonary alveolar, inducing a variety of respiratory diseases [11], such as cough,

dyspnea, chest pain [12], etc. It can also permeate into the bloodstream, elevating the risk of cardiovascular diseases within the circulatory system, such as hypertension and myocardial infarction. This can result in symptoms such as palpitations, dizziness, headaches, fatigue, and other physical discomforts [13,14]. These physical discomforts may reduce safe driving practices through subtle changes in driving behaviors since driving distracted is a significant cause of traffic accidents.

Smog pollution can also worsen the psychological state of individuals, resulting in more negative emotions [15]. Smog can exacerbate feelings of nervousness, worry, anxiety, and depression, along with other detrimental psychological effects. Additionally, it can contribute to heightened irritability, impulsivity, aggression, hatred, disgust, and other antagonistic negative emotions [16]. Furthermore, it may result in decreased cognitive function, slower thinking, and cognitive decline [17]. These negative psychological emotions, on the one hand, can reduce the driver's judgment and reaction velocity to the potential risks, and on the other hand, may also cause more "road rage", thus increasing the possibility of traffic accidents.

### 3. Methodology

#### 3.1. Model

We utilized the RDD to assess the impact of smog warning signals on traffic fatalities. The smog warning signal primarily relies on the PM2.5 concentration. When the concentration surpasses the warning threshold, the CEPA issues a warning signal, prompting drivers to adjust their driving behavior accordingly. Conversely, when the concentration is slightly below the threshold, no warning signal is issued, yet drivers may still be influenced by the warning. Therefore, we can regard the smog warning signal as an exogenous shock of a quasi-natural experiment, taking the samples with a PM2.5 concentration higher than the threshold as the experimental group and the samples lower than that concentration as the control group. In a small window around the concentration threshold, only the warning signal changes suddenly, while other factors remain continuous. If, in this smaller window, we observe that the number of traffic deaths in the experimental group also jumps suddenly before and after the warning signal compared to the control group, then the change in traffic deaths can be attributed to the effect of the warning signal.

We constructed independent RDD models based on three types of light smog, moderate smog, and extremely heavy smog early-warning signals.

$$Deaths_{it} = \alpha_0 + \alpha_1 Warning_{it}^J + \alpha_2 f\left(Concerntration_{it}^J\right) + \alpha_3 Control_{it} + \sum \chi_w + \gamma_t + \lambda_i + v_{it} \tag{1}$$

In Formula (1), $Deaths_{it}$ represents the number of traffic deaths in city $i$ at the time $t$ (date including the year, month, and day). $Warning_{it}^J$ is the policy treatment variable. When the city $i$ has a smog warning $J$ on day $t$, its value is equal to 1, otherwise its value is 0. $J$ is the level of smog warning, when J = 1, 2, and 3, it represents yellow, orange, and red warning signals, respectively. $Concerntration_{it}^J$ is the running variable of the model, which represents the PM2.5 concentration value of city $i$ on day $t$. $f\left(Concerntration_{it}^J\right)$ is the polynomial expression of the running variable. According to the suggestion of Gelman and Imbens, we conducted estimations in linear and quadratic polynomial forms. $Control_{it}$ are continuously changing meteorological factors, such as temperature and humidity, etc.; the discrete control variables $\chi_w$ represent different weather conditions and they are controlled using the high-dimensional fixed-effect method; $\gamma_t$ is a daily time fixed effect, $\lambda_i$ is a city fixed effect, and $v_{it}$ is the error term.

#### 3.2. Data

The traffic death data in this paper came from the Chinese Judicial Document Network (https://wenshu.court.gov.cn/), accessed on 22 April 2022. The Supreme People's Court of China stipulates that starting from 2014 the people's courts must post their effective judgments on the Chinese Judicial Document Network. For severe cases such as traffic

deaths, as long as the driver is at fault, the regional people's procuratorate will prosecute him for suspected "traffic accident crime", and the court will form a judgment document and record it in China Judgments online. We focused on the "traffic accident crime" judgment texts from 2014 to 2021 and downloaded them from the China Judgments online. We found that the number of judgment documents collected before 2016 was small, which means that samples are missing; at the same time, since the legal proceedings of traffic crime usually take several months, the judgment documents of traffic death cases in 2021 were not likely to be all concluded in the same year. Therefore, we decided to use a five-year study period from 2016 to 2020.

The text content of the court's judgment had a consistent logical structure, especially for cases with relatively simple circumstances such as "traffic accident crime". The content usually included the following four aspects in sequence: the first is the introduction of the demographic information of the driver who caused the accident and the judicial processes he experienced; the second is the prosecution's statement to the driver of the accident and its outcome, as well as the corresponding charges; the third is the defense statement of the driver who caused the accident as the defendant against the charges of the public prosecution agency; and the fourth is the statement of the court's decision. Therefore, the verdict text of "traffic accident crime" contained significant helpful information, which could fully meet the research needs of this paper. At the same time, the logically consistent text content also provided convenience for us to obtain structured research data.

This article used the text recognition and crawling functions of EXCEL and Python to obtain traffic death data from the text of the "traffic accident crime" judgment through the following steps. Firstly, we deleted the judgment documents from the second and third trials because the driver who caused the accident may have had an appeal, which would lead to multiple judgment documents in the same case, resulting in a large and inaccurate research sample. Secondly, we identified the text associated with "death" and deleted those "traffic crime" records that did not cause death, such as drunk driving and so on, to ensure that each sample was a case of traffic death. Thirdly, we extracted the number of people who died in the accident from the summary statement of the judgment. Fourthly, from the detailed process statement by the public prosecutor, we located and crawled the exact date and city of the accident, the characteristics of the driving vehicle (two or four wheels), and the road section characteristics (city or country, freeway or non-freeway). We also crawled the driver's population characteristics (sex, education, and age) from the driver's information. Finally, we calculated the daily traffic fatalities for 295 cities in China from 2016 to 2020.

Compared with existing traffic accident research datasets, our traffic crime data from the Chinese Judicial Document Network may have had three advantages. First, our data were a national overall sample. Much of the existing evidence on traffic crashes is based on regional samples, whereas we used broader overall data. Next, we selected the daily high-frequency data to facilitate the integration of weather and daily smog pollution data on the spatial and temporal dimensions, which was expected to improve the precision of the estimates. Finally, every criminal record contained detailed information such as the driver's population characteristics, the specific location of the collision, and the characteristics of the vehicle which could be used to conduct a detailed heterogeneity analysis to discover more about the effects.

### *3.3. Variables*

#### 3.3.1. Explained Variables

Our main explained variable was the daily traffic deaths in China's 295 cities from 2016 to 2020. From the documents, we collated the drivers' personal characteristics, vehicle types, and road locations. Based on the information extracted above, we classified driver-culpable traffic deaths into the following sorts to further study the different effects on different drivers, vehicles, and roads: man driver, woman driver, older driver (age $\geq$ 60 years old), young driver (age $\leq$ 35 years old), middle-aged driver (35 < age < 60 years old), driver

with less-education (under university), well-educated driver (university and above), four-wheeled vehicle, two-wheeled vehicle, city road, country road, freeway, and non-freeway.

### 3.3.2. Treatment Variable and Running Variable

The treatment variable "whether early-warning" represents whether the daily PM2.5 concentration reached the corresponding early-warning critical value. In this paper, it was assumed that when the daily PM2.5 concentration exceeded the critical threshold for early warning, it was considered a warning sample, and the corresponding value was set to 1. Otherwise, it was set to 0. The running variable was also expressed by the daily concentration of PM2.5, and the PM2.5 data were from the China Environmental Monitoring Station.

The effectiveness of our RDD strategies depended on the continuity of the running variables. If the running variables were not randomly and continuously distributed at the cutoffs, then the estimated effect may have been due to the running variables rather than the early-warning policy, making our RDD strategies unreliable. To resolve the above issue, we adopted the approach of Cattaneo, Jansson, and Ma [18] to conduct the DC−density test on the daily PM2.5 concentration of the running variables. As depicted in Figure 1, the density distributions of the running variables exhibited smooth and continuous patterns around the thresholds for the light smog, moderate smog, and extremely heavy smog early-warning signals. There were no evident "jumps" observed around the cutoff points, suggesting that the continuity assumption of the running variables held true.

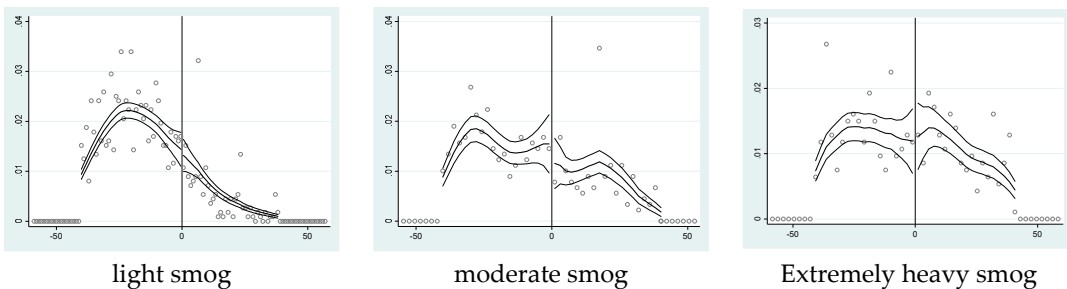

|  light smog  |  moderate smog  |  Extremely heavy smog  |

**Figure 1.** DC−density test of treatment variables in different smog level warnings.

### 3.3.3. Controlling Variables

Taking the daily meteorological factors related to smog pollution and traffic as controlling variables, including continuous and discrete variables, could not only improve the accuracy of the estimation but also prevent an estimation error as much as possible. The continuous variables were humidity and temperature. The discrete variables included eleven weather categories: sprinkle, middle rain, hard rain, thundery rain, sleet, slight snow, great snow, multi-cloud, clear day, dull day, and foggy; nine wind directions, eastern, northeast, southeast, northern, northwest, southwest, western, southern, and windless; and four wind velocity classifications, 0~3, 3~4, 4~5, and above 5. The descriptive statistics for the control variables are presented in Table 1. All the meteorological data were obtained from the China National Environmental Monitoring Centre.

**Table 1.** Summary statistics.

|  | N | Mean | S.D. | Min. | Max. |
|---|---|---|---|---|---|
| Total road deaths | 538,965 | 0.418 | 0.156 | 0 | 11 |
| PM2.5($\mu$g/m$^3$) | 538,965 | 46.654 | 30.142 | 0 | 1033 |
| Maximum temperature (°C) | 538,965 | 19.928 | 3.570 | −41 | 52 |
| Minimum temperature (°C) | 538,965 | 9.554 | 3.529 | −56 | 36 |
| Mean temperature (°C) | 538,965 | 14.654 | 3.684 | −20 | 38 |
| Humidity (%) | 538,965 | 67.424 | 18.245 | 3 | 100 |
| Weather categories | 538,965 | 3.812 | 2.380 | 1 | 11 |
| Wind direction | 538,965 | 3.543 | 2.645 | 1 | 9 |

**Table 1.** *Cont.*

|  | N | Mean | S.D. | Min. | Max. |
|---|---|---|---|---|---|
| Wind velocity class | 538,965 | 1.384 | 0.667 | 1 | 4 |
| *Deaths caused by different drivers* | | | | | |
| Male | 538,965 | 0.202 | 0.164 | 0 | 11 |
| Female | 538,965 | 0.215 | 0.135 | 0 | 5 |
| The elderly (above 60) | 538,965 | 0.072 | 0.059 | 0 | 4 |
| The young (below 35) | 538,965 | 0.179 | 0.107 | 0 | 11 |
| The middle-aged (from 36 to 59) | 538,965 | 0.165 | 0.096 | 0 | 6 |
| The less-educated (under university) | 538,965 | 0.281 | 0.173 | 0 | 11 |
| The well-educated (university and above) | 538,965 | 0.134 | 0.084 | 0 | 5 |
| *Deaths caused by different vehicles* | | | | | |
| Four-wheel | 538,965 | 0.212 | 0.169 | 0 | 11 |
| Two-wheel | 538,965 | 0.203 | 0.124 | 0 | 3 |
| *Deaths on different roads* | | | | | |
| City | 538,965 | 0.218 | 0.172 | 0 | 11 |
| Country | 538,965 | 0.198 | 0.114 | 0 | 7 |
| Freeway | 538,965 | 0.168 | 0.093 | 0 | 11 |
| Non-freeway | 538,965 | 0.247 | 0.128 | 0 | 6 |

Note: Table 1 provides a summary of the statistics of the dataset. The variable "N" represents the sample size, indicating the number of observations. "Mean" represents the average value of the samples, providing a measure of central tendency. "S.D." represents the standard deviation, indicating the dispersion or variability of the data points around the mean. "Min." refers to the minimum value observed in the samples, representing the lowest data point; similarly, "Max." refers to the maximum value observed, representing the highest data point in the samples.

## 4. Results

### 4.1. Basic Results

#### 4.1.1. Discontinuity Fitting Curves

To obtain a more intuitive understanding of the effect of smog early-warning signals on traffic deaths, we presented linear and quadratic trends in the number of traffic deaths before and after the three early warnings, with fitting curves as shown in Figures 2–4. Each point in the graph represents the number of traffic deaths at the corresponding PM2.5 concentration, and the solid line represents a non-parametric fit to the data; a jump in the number of daily traffic fatalities is evident after the light smog and moderate smog early-warning thresholds, while the jump at the extremely heavy smog early-warning threshold is less pronounced than at the light smog and moderate smog early-warning thresholds. From the point of scatter distribution, the light smog early warning is relatively concentrated, better than the moderate smog early-warning distribution; last is the extremely heavy smog early warning. The light smog and moderate smog early warnings are relatively stable in the trend change of the curve before and after the breakpoints, while the extremely heavy smog early warning is in the opposite direction.

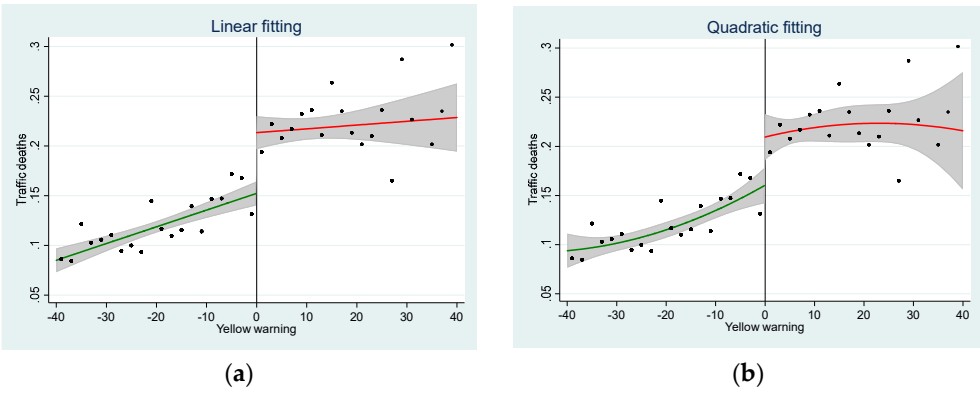

(a)　　　　　　　　　　　　　　　　　　　　(b)

**Figure 2.** The light smog early−warning fitting. (**a**) Linear and (**b**) quadratic.

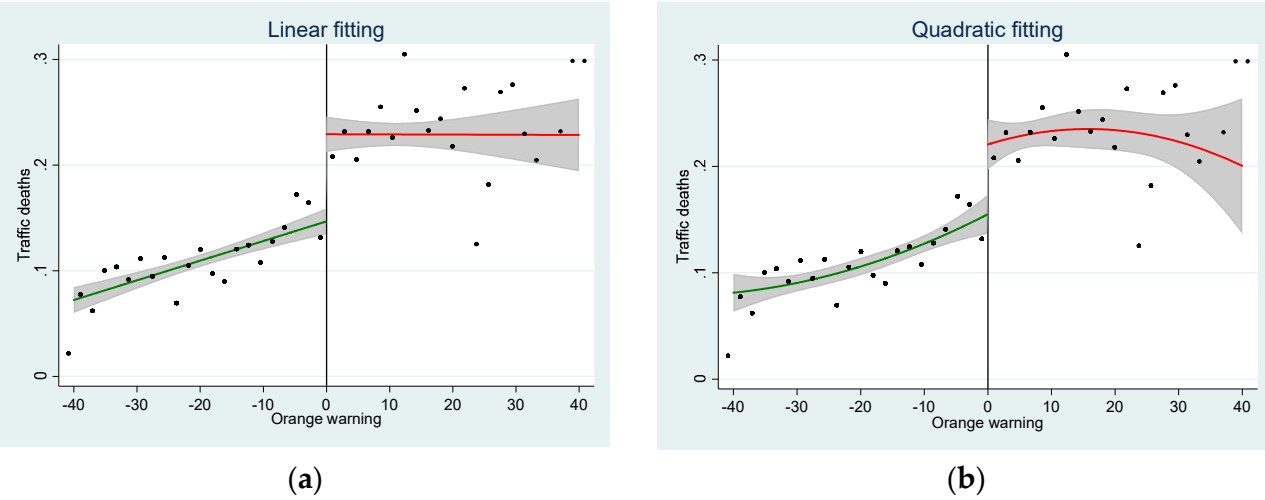

**Figure 3.** The moderate smog early−warning fitting. (**a**) Linear and (**b**) quadratic.

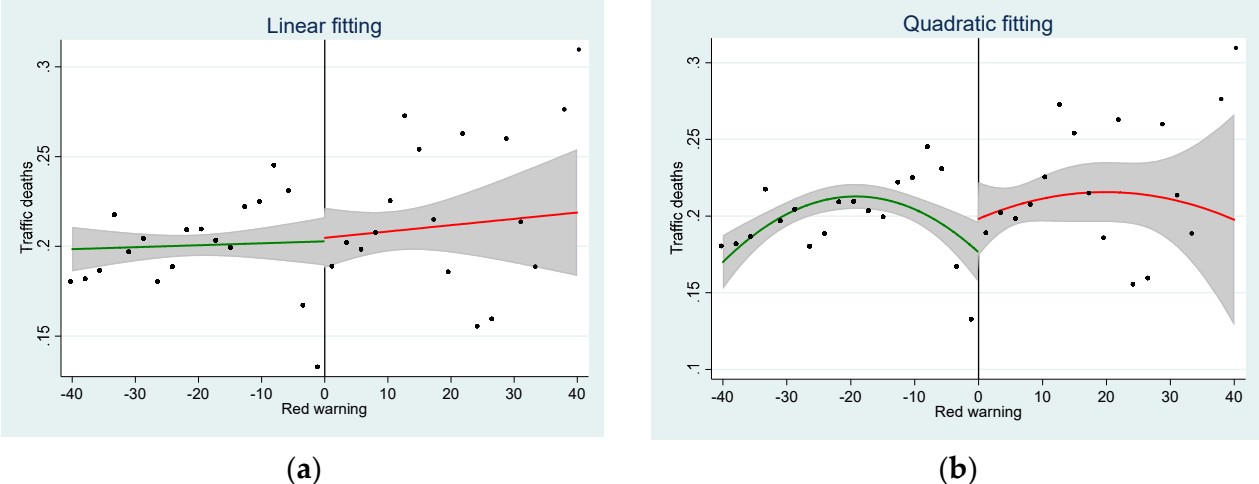

**Figure 4.** The extremely heavy smog early−warning fitting. (**a**) Linear and (**b**) quadratic.

Note: In Figure 2, ".1" represents "0.1" and ".15" represents "0.15". Similarly, in other figures, similar numbers should be interpreted accordingly.

### 4.1.2. Estimating Results

We used model (1) to conduct RDD estimation of the light smog, moderate smog, and extremely heavy smog early-warning signals on traffic deaths. In Table 2, the first and second columns illustrate the linear and quadratic regression results of the light smog early-warning signal, respectively. The coefficients of the early-warning policy before the treatment variable are significantly negative, signifying that the light smog early-warning signal will significantly reduce traffic deaths. Specifically, one light smog early-warning signal will reduce the total death traffic deaths by about 3.6% (0.015/0.418). Columns (3) and (4) are the regressions of the moderate smog early-warning signal. Similarly, its coefficients before the policy treatment variable are significantly negative, indicating that issuing a moderate smog early-warning signal will reduce the total traffic deaths by about 4.3% (0.018/0.418). Surprisingly, we find from columns (5) and (6) that the estimated coefficients for extremely heavy smog early-warning signals, although negative, are not significant, which means that an extremely heavy smog early-warning signal would not cause a reduction in traffic deaths.

**Table 2.** RDD estimations of smog early warnings on traffic deaths.

| | Light Smog | | Moderate Smog | | Extremely Heavy Smog | |
|---|---|---|---|---|---|---|
| | (1) | (2) | (3) | (4) | (5) | (6) |
| | Linear Model | Quadratic Model | Linear Model | Quadratic Model | Linear Model | Quadratic Model |
| Early-warning | −0.014 *** (0.004) | −0.015 *** (0.004) | −0.018 ** (0.009) | −0.018 ** (0.011) | −0.008 (0.010) | −0.009 (0.010) |
| Control | Y | Y | Y | Y | Y | Y |
| Weather fixed effects | Y | Y | Y | Y | Y | Y |
| Date fixed effect | Y | Y | Y | Y | Y | Y |
| City fixed effect | Y | Y | Y | Y | Y | Y |
| N | 72,574 | 72,574 | 36,454 | 36,454 | 8356 | 8356 |

Note: The standard errors are in brackets. ** $p < 0.05$, and *** $p < 0.01$.

The RDD results indicate that the light smog and moderate smog early warnings significantly reduce traffic accident fatalities, but the extremely heavy smog early-warning signals do not, which is consistent with the results shown by discontinuity fitting curves. There are two mechanisms that may determine an individual's response to smog pollution. The first is the self-awareness mechanism, that is, individuals autonomously perceive smog pollution through visual observation or other physiological methods, and actively take protective measures without the early-warning signal; the second is the early-warning signal mechanism, that is, individuals are less sensitive to the occurrence of smog pollution, and the issuance of the early-warning signal of smog makes them aware of the high degree of outdoor environmental pollution and take corresponding preventive measures. We believe that when PM2.5 concentrations are close to the light smog and moderate smog early-warning breakpoints, drivers often cannot directly perceive environmental smog pollution through sight lines. At this time, they need to receive the smog early-warning signal before making a behavioral response to drive carefully, that is, the self-recognition mechanism does not work while the early-warning signal mechanism works. When the concentration of PM2.5 is near the extremely heavy smog early-warning critical value, the driver will notice the smog pollution, and take countermeasures spontaneously. In this case, the extremely heavy smog early-warning signal will no longer serve as a reminder. We believe that the failure of the extremely heavy smog early warning is due to the self-awareness mechanism, not the early-warning signal mechanism.

*4.2. Robustness Tests*

4.2.1. Continuity Tests

We tested the continuity of PM2.5 concentration as a running variable in the 3.3 Variables section, and here we further test the continuity of other covariates. The covariates in this paper are continuous weather variables that affect traffic safety and smog pollution, mainly including maximum temperature, minimum temperature, mean temperature, and humidity. If these covariates show a significant jump at the cutoffs when the smog early warnings are issued, the effects of sudden "jumps" in traffic deaths could possibly be due to these jumping covariates other than the smog early-warning signal, thus resulting in the bias of our previous estimates.

We examined the continuity of covariates at cutoffs by plotting the fitted curves of PM2.5 concentrations to those covariates. Figures 5–7 show the results of the covariates test at the light smog, moderate smog, and extremely heavy smog early-warning policy cutoffs. It can be seen from the figure that each covariate shows a good continuity before and after the policy cutoffs and that there are no obvious jumps, indicating that the covariates satisfy the continuity assumption, which confirms the reliability of our RDD basic estimation results.

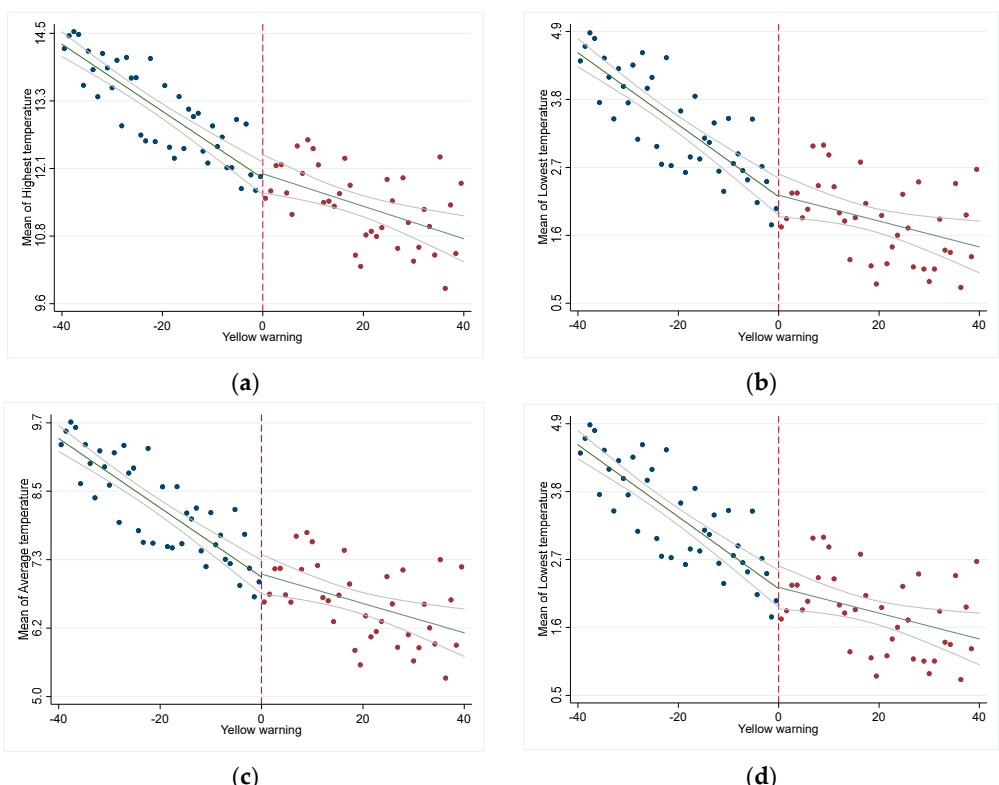

**Figure 5.** Continuity test at the light smog early−warning critical value. (**a**) Maximum temperature, (**b**) minimum temperature, (**c**) mean temperature, and (**d**) humidity.

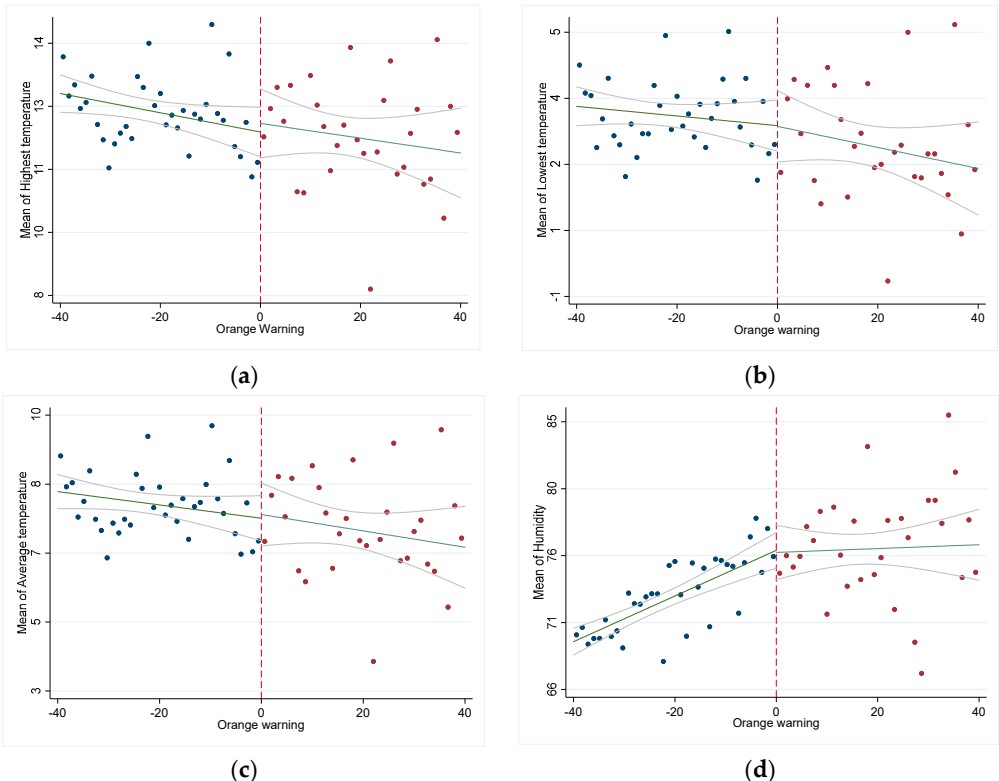

**Figure 6.** Continuity test at the moderate smog early−warning critical value. (**a**) Maximum temperature, (**b**) minimum temperature, (**c**) mean temperature, and (**d**) humidity.

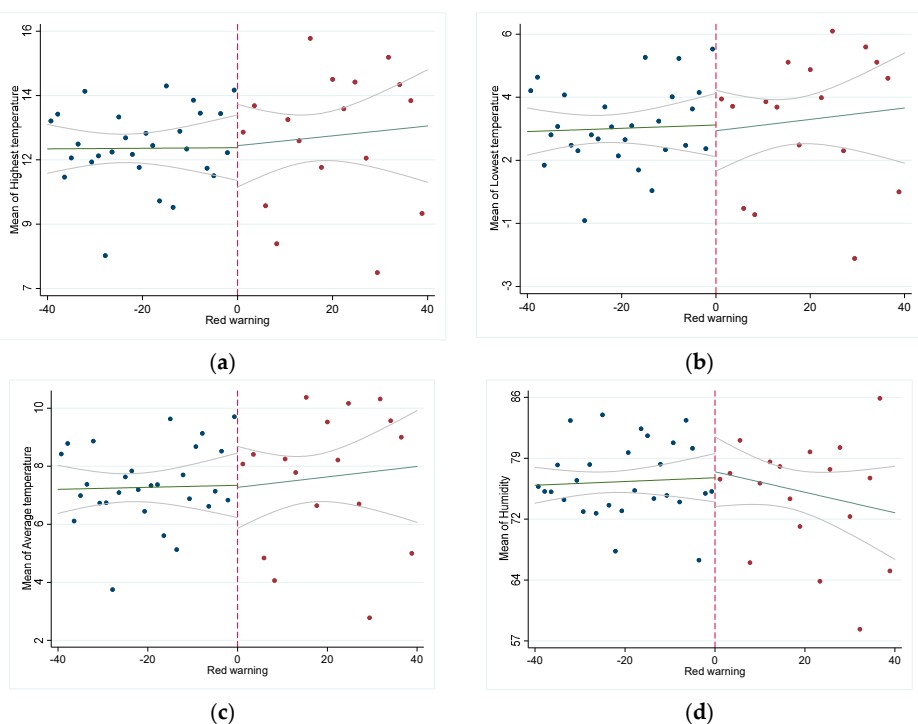

**Figure 7.** Continuity test at the extremely heavy smog early−warning critical value. (**a**) Maximum temperature, (**b**) minimum temperature, (**c**) mean temperature, (**d**) humidity.

### 4.2.2. Bandwidth Tests

The RDD estimation results are also affected by the bandwidth range [19]. In this section, we used parametric and non-parametric methods to test the sensitivity of bandwidth selections. First, we used the parametric method to test. The results are shown in Table 3. Based on the initial bandwidth {−40, 40} in the benchmark regressions, we adjusted the bandwidth to {−30, 30}, {−20, 20}, and {−10, 10} to re-estimate. As shown in Figure 3, the estimated coefficients of the light smog and moderate smog early warnings are still significantly negative in the new bandwidths, while the extremely heavy smog early warning is also insignificant. Regarding the value of the estimated coefficients, although the absolute value is slightly larger than the basic results, they are basically consistent with the previous results, which shows that our RDD estimation strategy does not depend on the selection of bandwidth.

**Table 3.** RDD results within different bandwidths.

|  | Light Smog | | | Moderate Smog | | | Extremely Heavy Smog | | |
|---|---|---|---|---|---|---|---|---|---|
|  | **B = 30** | **B = 20** | **B = 10** | **B = 30** | **B = 20** | **B = 10** | **B = 30** | **B = 20** | **B = 10** |
|  | (1) | (2) | (3) | (4) | (5) | (6) | (7) | (8) | (9) |
| Early warning | −0.016 *** | −0.017 *** | −0.018 *** | −0.020 ** | −0.022 ** | −0.022 ** | −0.009 | −0.006 | −0.006 |
|  | (0.002) | (0.003) | (0.003) | (0.011) | (0.011) | (0.012) | (0.007) | (0.008) | (0.009) |
| Control | Y | Y | Y | Y | Y | Y | Y | Y | Y |
| Weather fixed effects | Y | Y | Y | Y | Y | Y | Y | Y | Y |
| Date fixed effect | Y | Y | Y | Y | Y | Y | Y | Y | Y |
| City fixed effect | Y | Y | Y | Y | Y | Y | Y | Y | Y |
| N | 52,464 | 26,432 | 12750 | 24,436 | 15,285 | 10,242 | 6246 | 3524 | 1688 |

Note: The standard errors are in brackets. ** $p < 0.05$, and *** $p < 0.01$.

We further conducted a bandwidth test by non-parametric estimation. Based on the optimal bandwidth of the RDD non-parametric estimation, we estimated by sequentially expanding 10% of the optimal bandwidth in the range of 20–200% so as to test its sensitivity to the selections of bandwidth. As shown in Figure 8, the estimated results are mainly distributed between 0.013 and 0.015, with significance at the 5% confidence level, which is mostly consistent with the basic estimate and also convinces us that our benchmark parameter estimation results are robust.

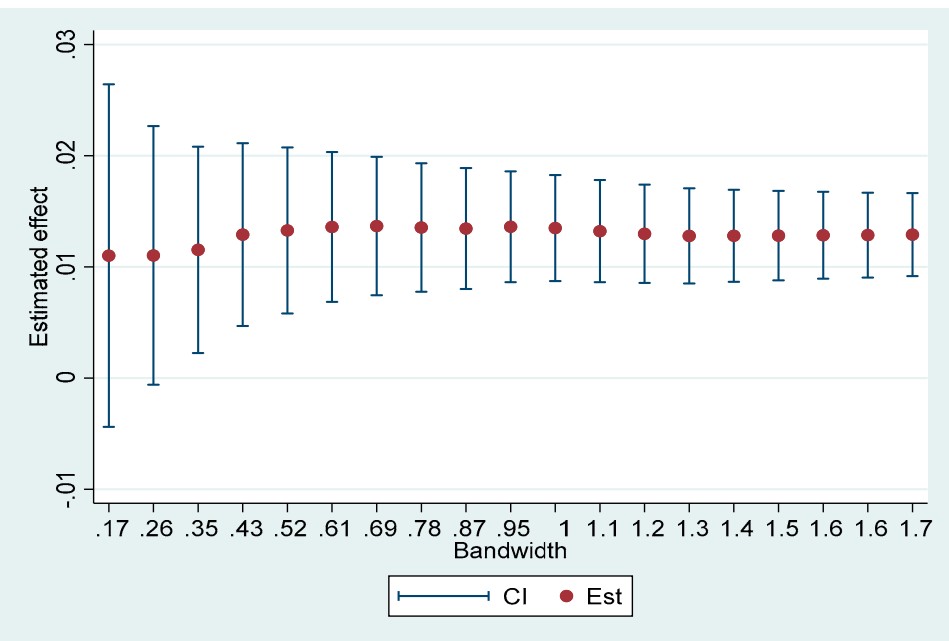

**Figure 8.** Non−parametric estimation of the bandwidth test.

*4.3. Heterogeneous Effects*

4.3.1. Driver Characteristics

As in Table 4, we first inspected the heterogeneous effects of smog early-warning signals on driver groups of different sexes, education levels, and ages.

The estimations of different sexes are in Panel A. In the light smog and moderate smog early-warning signals, the regression coefficients of the treatment variables were remarkably negative at 1% confidence levels for men and women drivers. Nonetheless, in the extremely heavy smog early-warning signal, only the man driver's coefficient was remarkable at the 10% confidence level, while the woman driver's coefficient was not remarkable. The estimated results specified that three early-warning signals affect men drivers significantly, but the extremely heavy smog early-warning signal does not affect women drivers significantly. One possible explanation for the differences is that women generally have more extraordinary sensitivity and insight to environmental pollution because of their mental and physical sensitivities. When the smog rises to the extremely heavy smog early-warning critical value, the self-awareness mechanism works more potently in women drivers, which means that women drivers more easily recognize potential risks and take preventive measures without relying on the extremely heavy smog early-warning signals. In addition, from the values of estimated coefficients, the absolute values of women in the light smog and moderate smog early-warning models were significantly larger than that of men, which indicates that women drivers are more likely than men to take precautions when receiving light smog and moderate smog early-warning signals. It was also correlated with their higher level of risk awareness in smog-polluted environments. When receiving the same pollution early-warning signal, women tend to respond more positively than men.

**Table 4.** Heterogeneous effects based on drivers' characteristics.

| | Panel A: Heterogeneous Effects on Sex | | | | | |
|---|---|---|---|---|---|---|
| | Light Smog | | Moderate Smog | | Extremely Heavy Smog | |
| | Man | Woman | Man | Woman | Man | Woman |
| Early warning | −0.005 *** (0.002) | −0.028 *** (0.009) | −0.008 *** (0.003) | −0.028 *** (0.010) | −0.008 * (0.005) | −0.009 (0.011) |
| | **Panel B: Heterogeneous Effects on Education** | | | | | |
| | Light smog | | Moderate smog | | Extremely heavy smog | |
| | Less-educated | Well-educated | Less-educated | Well-educated | Less-educated | Well-educated |
| Early warning | −0.011 *** (0.004) | −0.022 *** (0.006) | −0.012 *** (0.004) | −0.024 *** (0.004) | −0.009 (0.007) | −0.007 (0.007) |

| | Panel C: Heterogeneous Effects on Age | | | | | | | | |
|---|---|---|---|---|---|---|---|---|---|
| | Light smog | | | Moderate smog | | | Extremely heavy smog | | |
| | Young | Middle | Older | Young | Middle | Older | Young | Middle | Older |
| Early warning | −0.007 * (0.004) | −0.009 ** (0.004) | −0.024 *** (0.008) | −0.008 *** (0.003) | −0.017 *** (0.003) | −0.029 *** (0.004) | −0.006 ** (0.003) | −0.007 (0.007) | −0.005 (0.003) |
| Control | Y | Y | Y | Y | Y | Y | Y | Y | Y |
| Weather fixed effects | Y | Y | Y | Y | Y | Y | Y | Y | Y |
| Date fixed effect | Y | Y | Y | Y | Y | Y | Y | Y | Y |
| City fixed effect | Y | Y | Y | Y | Y | Y | Y | Y | Y |
| N | 72,574 | 72,574 | 72,574 | 36,454 | 36,454 | 36,454 | 8356 | 8356 | 8356 |

Note: The standard errors are in brackets. * $p < 0.1$, ** $p < 0.05$, and *** $p < 0.01$.

Panel B is the result of the heterogeneity estimates for different education levels. The estimated coefficients of the well-educated group and less-educated group were both negatively significant at the 1% level in the light smog and moderate smog early-warning models, while they were not significant in the extremely heavy smog early-warning. At the same time, in the light smog and moderate smog early-warning models, the absolute coefficient of drivers with less education was statistically smaller than that of well-educated drivers, indicating that early-warning signals have a greater impact on highly educated drivers. There are several possible explanations. Firstly, they usually have a broader knowledge base and better comprehension skills, allowing them to accurately understand the meaning and urgency of haze warning signals. Secondly, they possess higher risk awareness and decision-making abilities, enabling them to assess the threat of haze on driving safety more accurately and take appropriate measures to mitigate it. Lastly, they place a greater emphasis on health awareness and social responsibility, being willing to take proactive actions to protect their own and others' health [2].

The estimated coefficients of the age group are exhibited in Panel C. In the light smog and moderate smog early-warning models, the estimated results of the three age groups were negative; while in the extremely heavy smog early-warning model, only the coefficient of the young group was significant, which indicates that drivers of all age groups are susceptible to the light smog and moderate smog early-warning signals, while the extremely heavy smog early-warning signal only has an effect on young drivers. We believe that younger drivers may be more tolerant of environmental risks. When

confronting the possible harm caused by environmental risks, they need to receive external early-warning signals before taking preventive actions [20]. Comparing the estimated coefficient values of the three groups, it was found that elderly people had the largest absolute value of the coefficient, implying that elderly drivers are most affected when light smog and moderate smog early-warning signals are issued. Elderly drivers generally have more fragile and susceptible health conditions compared to younger drivers, making them more vulnerable to the impact of external environmental factors. Hazy weather can cause a decline in air quality, with harmful particulate matter and pollutants posing a greater threat to health. Elderly drivers may be more susceptible to the negative effects of hazy weather due to respiratory, cardiovascular, or other health issues, thus making them more sensitive to warning signals. Additionally, elderly drivers generally possess a strong awareness of safety and risk perception, prioritizing their own and others' safety. In hazy weather, reduced visibility and slippery roads increase the risk of traffic accidents. Due to their heightened sensitivity to driving risks, elderly drivers may be more attuned to the driving safety threats posed by light or moderate haze and therefore more vigilant towards warning signals [21].

### 4.3.2. Vehicle Types

Different types of vehicles will expose drivers to different risks of air pollution. For example, four-wheeled vehicles can usually close windows to lower drivers' direct exposure to smog pollution, while drivers of two-wheeled vehicles are completely exposed. Our results, as shown in Table 5, illustrate the effect of smog early-warning signals on four-wheeled vehicles and two-wheeled vehicles.

**Table 5.** Heterogeneous effects based on vehicle types.

| | Light Smog | | Moderate Smog | | Extremely Heavy Smog | |
| --- | --- | --- | --- | --- | --- | --- |
| | **2-Wheel** | **4-Wheel** | **2-Wheel** | **4-Wheel** | **2-Wheel** | **4-Wheel** |
| Early warning | −0.024 *** (0.004) | −0.007 *** (0.003) | −0.027 *** (0.008) | −0.012 ** (0.006) | −0.013 * (0.006) | −0.008 (0.009) |
| Control | Y | Y | Y | Y | Y | Y |
| Weather fixed effects | Y | Y | Y | Y | Y | Y |
| Date fixed effect | Y | Y | Y | Y | Y | Y |
| City fixed effect | Y | Y | Y | Y | Y | Y |
| N | 72,574 | 72,574 | 36,454 | 36,454 | 8356 | 8356 |

Note: The standard errors are in brackets. * $p < 0.1$, ** $p < 0.05$, and *** $p < 0.01$.

We discovered that in the light smog and moderate smog early-warning models, the estimated coefficients of the two-wheeled and four-wheeled vehicles are significantly negative, while in the extremely heavy smog early-warning model, only the coefficients for the two-wheeled vehicles are significantly negative, indicating that the light smog and moderate smog early-warnings affect both types of vehicles while the extremely heavy smog early-warning affect only two-wheeled vehicles significantly. We further compared the estimated coefficient values of the two types of vehicles and found that the estimated coefficient of a two-wheeled is statistically higher than that of a four-wheeled one, demonstrating that the early warning signal has a more evident effect on a two-wheeled vehicle than on a four-wheeled vehicle. We believe that two-wheeled vehicle drivers, who are in direct exposure to smog, are more easily subjected to physical and mental damage [22]. Compared to the four-wheeled drivers, the early warnings may decrease the interference of smog pollution on two-wheeled vehicle driving behavior.

### 4.3.3. Road Locations

We further investigated the possible changes in traffic deaths due to the issuance of smog early-warning signals at different road sections, such as city and country roads, freeways, and non-freeways, as shown in Table 6.

**Table 6.** Heterogeneous effects based on road locations.

| | Panel A: Heterogeneous Effects on City and Country Roads | | | | | |
| --- | --- | --- | --- | --- | --- | --- |
| | Light Smog | | Moderate Smog | | Extremely Heavy Smog | |
| | City | Country | City | Country | City | Country |
| Early warning | −0.012 *** (0.005) | −0.018 * (0.010) | −0.015 *** (0.006) | −0.018 *** (0.006) | −0.006 (0.008) | −0.013 ** (0.006) |
| | Panel B: Heterogeneous Effects on Freeway and Non-freeway Roads | | | | | |
| | Light smog | | Moderate smog | | Extremely heavy smog | |
| | Freeway | Non-freeway | Freeway | Non-freeway | Freeway | Non-freeway |
| Early warning | −0.018 *** (0.004) | −0.014 *** (0.004) | −0.023 *** (0.003) | −0.017 ** (0.006) | −0.013 ** (0.006) | −0.003 (0.008) |
| Control | Y | Y | Y | Y | Y | Y |
| Weather fixed effects | Y | Y | Y | Y | Y | Y |
| Date fixed effect | Y | Y | Y | Y | Y | Y |
| City fixed effect | Y | Y | Y | Y | Y | Y |
| N | 72,574 | 72,574 | 36,454 | 36,454 | 8356 | 8356 |

Note: The standard errors are in brackets. * $p < 0.1$, ** $p < 0.05$, and *** $p < 0.01$.

Panel A illustrates the regression results for city and country roads. In the light smog and moderate smog early-warning models, all the regression coefficients were remarkably negative, showing that the light smog and moderate smog early-warning signals do reduce road traffic deaths in city and country areas; in the extremely heavy smog early-warning model, although the regression coefficient of the city road did not pass the significance test, the regression coefficient of the country road is significantly negative at the 5% level, indicating that the extremely heavy smog early-warning signal only significantly affects country roads. Moreover, the absolute coefficient value of country roads was larger than that of city roads, indicating that compared with city roads, all the early warnings showed a more evident preventive effect on traffic deaths on country roads. This may be because drivers on country roads are generally less educated and more tolerant of pollution risks [23], consequently relying more on early-warning signs. Furthermore, compared to urban areas, rural areas typically have limited rescue and medical resources. In the event of a traffic accident during hazy weather, it may be more challenging for rural areas to provide timely emergency rescue and medical treatment, resulting in more severe consequences of accidents. Therefore, the issuance of smog warning signals has a more significant preventive effect on traffic fatalities in rural road areas. It can prompt drivers to be more cautious and attentive to traffic safety, thereby reducing the occurrence rate of accidents and the risk of fatalities.

Panel B is an estimate for both freeway and non-freeway roads. We found that the three early warnings reflected apparent effects on the prevention of freeway traffic death, but the extremely heavy smog early-warning alert had no remarkable effect on the prevention of non-freeway traffic death. Furthermore, the estimated absolute value of the coefficients for a freeway was significantly larger than those for a non-freeway. The possible explanation is that motorway traffic is faster and drivers tend to drive more cautiously, and when drivers on the highway receive an early-warning signal, they change their driving behavior more quickly to prevent smog from interfering. Another possible mechanism is that increased smog pollution may reduce highway visibility, leading to a reduction in the number of

vehicles on the highway, thereby reducing traffic accidents. Unfortunately, we do not have access to detailed daily highway driving data to empirically test this possibility. However, in our RDD strategy, we believe that in a small interval near the smog early-warning cutoff, a slight increase in the PM2.5 concentration value will not significantly change the highway visibility, and thus will not cause a significant reduction in the number of vehicles. In this sense, our RDD strategy is able to exclude this possible mechanism.

## 5. Discussion

This paper assesses the impact of an early-warning environmental policy rather than a regulatory one. The purpose of a "regulatory" environmental policy is to motivate polluters to reduce pollution emissions, while the goal of an "early-warning" environmental policy is to remind the public to prevent the harm caused by pollution. The former emphasizes reducing pollution emissions, while the latter highlights adapting to environmental changes. To deal with the deterioration of the ecological environment, such as greenhouse gas emissions, both "reduction" and "adaptation" are indispensable ways to ensure the sustainable development of mankind. Those two together form a combined policy system to deal with changes in the ecological environment, and we should not favor one over the other. However, the existing literature focuses on the roles of "regulatory" environmental policies, such as environmental laws and regulations [24,25], administrative rules and regulations, market-based emissions trading, the environmental protection tax mechanism [2], etc. In contrast, the "early-warning" environmental policy has not been given enough attention. Environmental early-warning policies can also play an essential role in promoting sustainable economic and social development, just as this paper reveals that smog warning signals will significantly reduce traffic fatalities. Future research on environmental policy assessment should pay attention to adaptive policies such as early-warning policies other than just regulatory ones.

There is much literature on traffic safety warnings, such as fog warnings, high-temperature warnings, rainstorm warnings, typhoon warnings [26–28], etc., and signals play a positive role in preventing traffic accidents. However, these warning signals are for extreme weather, not environmental factors. Unfavorable weather conditions can reduce traffic visibility or road adhesion, decreasing the driver's ability to perceive the surrounding environment and handle the vehicle well, resulting in more traffic accidents. Compared with the explicit impact of meteorological factors on traffic safety, the effects of smog pollution on traffic deaths are relatively implicit, which is more likely to cause traffic deaths by worsening drivers' physical and psychological states rather than their visibility. This paper expands the research on early warnings for traffic safety from meteorological factors to environmental factors.

## 6. Conclusions

To the best of our knowledge, this paper is the first to assess the impact of the environmental smog warning policy on traffic deaths. We obtained almost a whole sample of daily city traffic death data from 2016 to 2020 in China from the official China Judgment Online. Based on the PM2.5 concentration thresholds for warning signals, we constructed a rigorous RDD strategy to identify the smog warning policy and adopted the high-dimensional fixed-effect method to eliminate the interference of various meteorological factors in order to accurately estimate the positive effects of smog warning signals on traffic deaths.

This paper focuses on the impact of an environmental early-warning policy on traffic safety, which may enrich the research on environmental policies from the perspective of "warning" rather than "regulatory", and expands the scope of environmental policy into traffic safety, which is useful to policy-making that is both related to smog pollution regulation and transportation safety. We suggest that the traffic department should work with the meteorological department and environmental department to establish a joint meteorological–environmental traffic warning system, involving both extreme weather factors and environmental pollution factors, so as to reduce the number of traffic accidents

on a larger scope. Considering the significant impact of yellow and orange haze warning signals on reducing traffic accident fatalities, the government can more extensively utilize these warning signals in traffic management. These signals serve to alert drivers to road conditions and prompt them to take corresponding preventive measures, even when haze does not significantly affect visibility, thereby lowering the likelihood of traffic accidents. According to research findings, female drivers, drivers with higher education levels, elderly drivers, motorcycle riders, and drivers traveling on rural roads and highways are more susceptible to the influence of haze warning signals and exhibit corresponding preventive driving behavior. We recommend that the government develop targeted educational and training programs based on these characteristics to enhance awareness of haze warning signals among other demographic groups and encourage them to adopt proactive driving behavior. Future research on environmental policies can pay more attention to the "adaptive" policies, such as the smog warning policy, and can also learn from the RDD strategy constructed in this paper in terms of methodology.

This article also has some limitations. Firstly, the findings of the study focusing on traffic deaths in China may not be generalizable to other countries or regions with different traffic patterns, environmental policies, and cultural contexts. Replicating the study in diverse settings would be necessary to validate the results. Secondly, although the paper uses a rigorous RDD strategy to identify the impact of smog warning signals on traffic deaths, it is important to note that causality cannot be definitively established. There may be unobserved confounding factors that affect both the implementation of smog warning signals and traffic deaths which could introduce bias to the estimated effects. Lastly, while the paper conducts a heterogeneity analysis based on driver characteristics, vehicle types, and road locations to examine the differential effects of smog warning signals, the external validity of these findings may be limited. The study also does not explore other relevant factors that could influence the observed heterogeneity, such as regional differences in pollution levels or driver behaviors.

**Author Contributions:** Conceptualization, J.G. and C.Y.; methodology, C.Y.; validation, H.X., L.H. and Z.L.; formal analysis, L.H.; investigation, H.X.; resources, J.G.; data curation, H.X.; writing—original draft preparation, J.G.; writing—review and editing, J.G. and Z.L.; visualization, C.Y.; supervision, J.G.; project administration, J.G.; funding acquisition, J.G. All authors have read and agreed to the published version of the manuscript.

**Funding:** This paper is financially supported by the National Natural Science Foundation of China (Grant No. 72004170), the National Outstanding Youth Science Fund Project of China (Grant No. 71725007), the Chinese National Funding of Social Science (Grant No. 19ZDA083), and the Fundamental Research Funds for the Central Universities (Grant No. 2020SK040).

**Institutional Review Board Statement:** Not applicable.

**Informed Consent Statement:** Not applicable.

**Data Availability Statement:** The data presented in this study are openly available in Chinese Judicial Document Net-work (https://wenshu.court.gov.cn/), accessed on 22 April 2022.

**Conflicts of Interest:** The authors declare no conflict of interest.

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
