# Peer review of "Assessing the Effects of Environmental Smog Warning Policy on Preventing Traffic Deaths Based on RDD Strategy"

_atmosphere, doi:10.3390/atmos14061043_

Round 1
Reviewer 1 Report
This study aims to assess the impacts of environmental smog early-warning signals on road traffic deaths. For the scope, the daily traffic death data from 2016 to 2020 in 295 Chinese cities were used, and a rigorous Regression Discontinuity Design (RDD) strategy was constructed to identity the causality, adopting the high-dimensional fix effect method to deal with the interference of meteorological factors.
The paper is well organized. The issue dealt with is interesting, but major revisions are needed.
Point 1: The literature review reported in Section 2 should be supported by further literature studies to better identify the gap of knowledge.
Point 2: The size of the data sample considered was not indicated.
Point 3: The Figures are too small to be analysed.
Point 4: Table 1 should be better commented on. What does N indicate?
Point 5: The results should be compared with those found by other authors in order to demonstrate their reliability.
Point 6: The authors consider traffic deaths that occurred in five years (2016-2020), but it is unclear whether and how temporal correlation (data collected on the same section over successive time periods) was considered. In this respect, temporal correlation provides insight into the time trend of accidents over time. More details on temporal correlation can be found in:
· Caliendo, C.; Guida, M.; Postiglione, F.; Russo, I. A Bayesian bivariate hierarchical model with correlated parameters for the analysis of road crashes in Italian tunnels. Stat. Methods. Appl. 2022, 31, 109–131.
· Caliendo, C.; De Guglielmo, M.L.; Russo, I. Analysis of crash frequency in motorway tunnels based on a correlated random-parameters approach. Tunn. Undergr. Space Technol. 2019, 85, 243–251.
· Zeng, Q.; Guo, Q.; Wong, S.C.; Wen, H.; Huang. H.; Pei, X. Jointly modeling area-level crash rates by severity: a Bayesian multivariate random-parameters spatio-temporal Tobit regression. Transportmetrica A 2019, 15(2), 1867–1884.
Point 7: The limitations of the paper should be commented on.
Reviewer 2 Report
The study is about assessing the impacts of environmental smog early-warning signals on road traffic deaths in 295 cities in China.
The manuscript is satisfactorily-written, and the research is topical and timely, given the increasing road deaths worldwide. The authors have employed the Regression Discontinuity Design (RDD) approach to identifying causality and adopted the high-dimensional fix effect method to deal with the interference of meteorological factors.
The introduction section provides the research context and justifies the need for the study; however, it has unnecessary repetition of information provided in the abstract.
The literature review section needs to be strengthened by adding more recent and relevant studies that investigate the relationship between smog warning signals and road deaths in both developing and developed nations. This section lacks the necessary and relevant literature to support the authors' arguments.
The methodology section gives an appropriate level of detail on the modelling approaches and data collection methods adopted in the study.
The results of the modelling exercise need to be explained in a bit more detail, along with a comparison between the results of this study and those of other similar studies. Also, it must be made clear which shortcomings of the previous studies have been overcome by this study.
The conclusion section need not summarise the study but instead highlight the academic and practical contributions and applications of the study.
The writing quality needs to be improved moderately, as the authors tend to write complex and grammatically incorrect statements, which sometimes lack sense. Please see other comments made in the attached PDF document.

The study is about assessing the impacts of environmental smog early-warning signals on road traffic deaths in 295 cities in China.
The manuscript is satisfactorily-written, and the research is topical and timely, given the increasing road deaths worldwide. The authors have employed the Regression Discontinuity Design (RDD) approach to identifying causality and adopted the high-dimensional fix effect method to deal with the interference of meteorological factors.
The introduction section provides the research context and justifies the need for the study; however, it has unnecessary repetition of information provided in the abstract.
The literature review section needs to be strengthened by adding more recent and relevant studies that investigate the relationship between smog warning signals and road deaths in both developing and developed nations. This section lacks the necessary and relevant literature to support the authors' arguments.
The methodology section gives an appropriate level of detail on the modelling approaches and data collection methods adopted in the study.
The results of the modelling exercise need to be explained in a bit more detail, along with a comparison between the results of this study and those of other similar studies. Also, it must be made clear which shortcomings of the previous studies have been overcome by this study.
The conclusion section need not summarise the study but instead highlight the academic and practical contributions and applications of the study.
The writing quality needs to be improved moderately, as the authors tend to write complex and grammatically incorrect statements, which sometimes lack sense. Please see other comments made in the attached PDF document.
Round 2
Reviewer 1 Report
The paper can be accepted in its current form.